# Prevalence of Depression and Predictors of Discharge to a Psychiatric Hospital in Young People with Hospital-Treated Deliberate Self-Poisoning at an Australian Sentinel Unit

**DOI:** 10.3390/ijerph192315753

**Published:** 2022-11-26

**Authors:** Anitha Dani, Srilaxmi Balachandran, Katie McGill, Ian Whyte, Greg Carter

**Affiliations:** 1Child and Adolescent Mental Health Service, Hunter New England Mental Health Service, Newcastle, NSW 2302, Australia; 2School of Medicine and Public Health, University of Newcastle, Callaghan, NSW 2308, Australia; 3Research Evaluation and Dissemination (MH-READ), Hunter New England Mental Health Service, Newcastle, NSW 2298, Australia; 4Calvary Mater Newcastle Hospital, Waratah, NSW 2298, Australia

**Keywords:** self-poisoning, self-harm, suicide attempt, depression, psychiatric hospitalisation

## Abstract

Objective: Hospital treated deliberate self-poisoning is common in young people. Internationally, estimates of rates of depression in this population are very wide (14.6% to 88%). The aims of this study were to determine the prevalence of depression and the independent predictors of referral for psychiatric hospitalisation in young people (aged 16 to 25 years) following an index episode of hospital treated deliberate self-poisoning. Method: A retrospective cohort study design (*n* = 1410), with data drawn from a population-based clinical case register. Unadjusted and adjusted estimates of predictors of referral for psychiatric admission (after-care) used logistic regression models. Results: Prevalence of any depression diagnosis was 35.5% (n = 500); and 25.4% (n = 358) were referred for a psychiatric admission. The adjusted estimates for predictors of psychiatric inpatient referral were: high suicidal level (OR 118.21: CI 95% 63.23–220.99), low/moderate suicidal level (14.27: 9.38–21.72), any depression (2.88: 1.97–4.22), any psychosis (4.06; 1.15–14.36), older age (1.12: 1.04–1.21), and number of support people (0.88: 0.78–0.98). Conclusion: Depression was diagnosed in more than a third and was an independent predictor of psychiatric inpatient referral, so service providers need to account for this level of need in the provision of assessment and after-care services. Evidence-based guidelines for psychiatric inpatient after-care for deliberate self-poisoning and/or depression in young people are limited. Our explanatory model included suicidal level, depression, psychosis, older age, and available support persons, suggesting that the treating clinicians were making these discharge decisions for admission in keeping with those limited guidelines, although the balance of benefits and harms of psychiatric hospitalisation are not established. Future research examining patient experiences, effectiveness of psychiatric hospitalisation, and alternatives to hospitalisation is warranted.

## 1. Introduction

In Australia, all-age hospital-treated deliberate self-harm (injury and poisoning) is common. Presentations are mostly via the Emergency Department (ED), however, the estimates of prevalence are imprecise because of a reliance on institutional data that are acknowledged to underestimate the true rate [1]. In a five-year retrospective study of 115 Emergency Departments in NSW, there were 16,215 self-harm—including thoughts of self-harm and threats of self-harm, (mean age 28.4 years) and 8638 intentional self-poisoning (mean age 32.7 years) events [2]. Hospital-treated deliberate self-harm of adolescents and young adults is also common and costly. A 10-year retrospective, Australia-wide study of intentional injury hospitalisations of children aged 16 years or less, reported 18,223 self-harm (injury and poisoning) hospitalisations, with an event rate of 59.8/100,000 with an estimated treatment cost of AUS $64 million; with self-poisoning as the most common method of self-harm [3]. A study of NSW Emergency Departments over 5 years, reported a self-injury (including thoughts and threats of self-injury) peak annual event rate of 180/100,000 for self-injury and thoughts of self-injury and 75/100,000 for self-poisoning in the 10–19 years age group [2]; and a peak annual event rate 80/100,000 for self-injury and thoughts of self-injury and 40/100,000 for self-poisoning in the 20–39 years age group [2].

There is also uncertainty about whether there is a real or artefactual increase in hospital-treated deliberate self-harm in older teenagers and young adults in recent years. The Australian Institute of Health and Welfare has reported an increase in the age-specific rate of hospitalisations for intentional self-harm in young people over 10 years; (an increased annual event rate of 246 to 354/100,000 for 15–19 years and 220 to 252/100,000 for 20–24 years [4]. Conversely, a sentinel unit in Newcastle, Australia, using a population-based case register, has demonstrated no change over a ten-year period for young persons (15–24 years); with reported annual event rates for deliberate self-poisoning of 444/100,000 for females and 166/100,000 for males [5].

Hospital-treated deliberate self-poisoning in adolescent or young adult populations have high rates of psychiatric diagnoses [6,7]. In adult hospital-treated deliberate self-poisoning populations, the reported prevalence of “depression” (variously defined or classified) ranged from 5% in Denmark (Affective Disorder) [8] to 63.8% in Sri Lanka (“depressed”) [9]; and in young people 14.6% in Romania (Major Depression-severe), [10], 25% in Australia (Any Affective Disorder) [11], 67% in UK (Major Depression) [7] to 88% in Israel (“transient depression”) [12]. The wide variation in reported prevalence can be attributed in part to heterogeneity in study design (case series, cross sectional, case–control, cohort), sampling frames, diagnostic instruments (clinical review, standardized instrument, special questionnaire formulated for the study), diagnoses reported (individual disorders with or without severity, class of disorder such as any affective or mood disorders) and the classification systems used (ICD, DSM, unspecified).

One of the important and controversial after-care options for hospital-treated deliberate self-poisoning (or self-harm) is psychiatric hospitalisation. Internationally, the rates of referral (or admission) to a psychiatric unit for all-age populations following hospital-treated deliberate self-harm vary from 8.9% in Scotland [13] to 66.3% in Switzerland [14]. The variation in rates might be explained in part by the available resources (e.g., psychiatric bed-base or community-based mental health service availability, short stay unit in the Emergency Department of the general hospital), local legislation (e.g., Mental Health Act) and local clinical practices. From a limited evidence base for young people with hospital-treated self-harm, young adult age groups may be more likely to be directed to psychiatric inpatient treatment as after-care than adolescents; the UK reported a 2.6% rate of psychiatric hospitalisation in 10–18 years [15], 2.5% in 10–19 years [16], 7.8% in an all-age study [17], while Australian studies reported a rate of 15.6% in 12–17 years [18], and 20.1% in those under 25 years [19], with greater proportions seen in the Australian than UK populations. Despite limited research evaluating efficacy for reducing suicide [20] or other outcomes [21], and increasing awareness of potential harms [22,23], psychiatric hospitalisation is commonly used in the after-care of hospital-treated deliberate self-poisoning.

Although we have some international point estimates of rates of referral to psychiatric hospital for inpatient care, we know very little about the factors that might be associated with this important clinical decision, especially in young people. In an Australian study of all-age hospital treated deliberate self-poisoning, psychiatric hospitalisation on discharge from the general hospital was independently predicted by multiple patient (older age, mood or psychotic disorders, suicidal ideation or plan), social (marital status, homelessness, unemployment), and service characteristics (previous self-harm, psychiatric inpatient treatment within 12 months, earlier psychiatric inpatient treatment, lower clinician experience, after hours presentation) [19]. The pattern of independent predictors for psychiatric hospitalisation after deliberate self-poisoning in adolescents and young adults is not known.

### 1.1. Aims

The aims of the study, in a population of 16–25 years old presenting to Emergency Department after a deliberate self-poisoning episode, were:Estimate the prevalence of any Depression by clinician diagnosisEstimate the unadjusted magnitude of the association of any Depression diagnosis with referral for an inpatient psychiatric admission on discharge from the general hospitalEstimate the independent association of Depression diagnoses with referral for an inpatient psychiatric admission after adjustment for other key variablesDevelop an exploratory, explanatory model of independent predictive factors best explaining discharge to the psychiatric hospital.

### 1.2. Study Hypothesis

There will be no statistically significant association of a clinical diagnosis of any Depression with the discharge destination (referral for psychiatric inpatient care) in a clinical population of 16–25 years old participants, for an index episode of hospital-treated deliberate self-poisoning.

## 2. Materials and Methods

### 2.1. Setting

The study was conducted in the Calvary Mater Newcastle hospital in New South Wales, Australia, located in a health district covering metropolitan and regional areas. The health district encompasses a major teaching hospital, several large regional centres, smaller rural centres, and remote communities within its borders. It is estimated that this health district provides services to over 934,000 people of all ages and over 111,000 (11.8%) people aged 15 to 24 years.

### 2.2. Procedures

The Calvary Mater Newcastle provided a regional service for self-poisoning via the Hunter Area Toxicology Service (HATS). The model of care and associated clinical case register has been described in detail elsewhere [24]. Briefly, all poisoning presentations to Emergency Department were admitted under the clinical care of the attending clinical toxicologist and all deliberate self-poisoning patients received a mental health assessment from the Consultation-Liaison service during business hours (i.e., Monday–Friday 8:30 a.m. to 5 p.m.) or via the Hunter New England Mental Health after-hours service. The mental health assessment was conducted by psychiatry registrars, clinical nurse consultants or psychiatrists. Medical and psychiatric staff used a preformatted clinical record sheet to retrospectively collect information (toxicological, clinical, and psychiatric) on all patients. Diagnoses were made in accordance to the prevailing version of the Diagnostic and Statistical Manual for Mental Disorders (DSM-IV and DSM-5) at the time and were based on clinical assessments (patient interview, review of case notes, collateral information from patient’s family, close community supports, and treatment services). All cases were reviewed weekly, and the diagnoses confirmed at a multidisciplinary meeting. Data was prospectively entered into a relational database (an established clinical case register) by trained clinical staff that were blinded to any study aims.

### 2.3. Study Design

A retrospective cohort study design was used. Participants were chosen for the study if they had an episode of hospital-treated deliberate self-poisoning (DSP) at HATS during a ten-year period and were aged 16–25 years (inception rule). Only the first presentation in the period was used as the index episode for all data extraction and analyses, to avoid a lack of independence between observations. Primary outcome of the cohort design was discharge to the psychiatric hospital, with exposures including a number of participant demographic and clinical characteristics.

Exclusion criteria:accidental, occupational or iatrogenic poisonings,recreational use or chronic misuse poisonings,cases without a mental health assessment.

### 2.4. Variables

#### 2.4.1. Demographic Characteristics

Categorical: Age was stratified into two levels; adolescents aged 16–19 years and young adults aged 20–25 years. Other categorical variables were: gender identity (male, female), cultural background (Indigenous or non-Indigenous), marital status (single/never married, married/defacto, no longer partnered), employment status (employed or student, not working or studying, and unknown), highest level of education (primary school, high school, tertiary, and unknown), housing (secure, insecure, and unknown), referral area (primary, secondary, and not in referral area), Socio-Economic Indexes for Areas (SEIFA) (quintile 1, quintile 2–4, and quintile 5) [25]; and continuous variables: age at presentation, length of stay in the Emergency Department (reported in hours).

#### 2.4.2. Clinical Characteristics

Categorical: We used DSM-IV [26] and DSM-5 [27] major diagnostic categories (see below), and a three-level variable for suicidal level at the time of mental health assessment (none—no suicidal thoughts or plan; low/moderate—some suicidal thoughts but no plan; high—intense suicidal thoughts and/or plan; and unknown), any previous self-harm, previous psychiatric service in the prior 12 months (no contact, outpatient only, inpatient), significant life events (loss of someone close due to death or separation, newly diagnosed pregnancy, loss of pregnancy, loss of job, homelessness) in the 1 month prior to presentation (none, 1, 2 or more).

Continuous: Number of individual DSM diagnoses recorded, number of available support persons, number of life events in the 1 month before hospital presentation.

Referral for psychiatric hospitalisation after hospital-treated DSP episode (dependent variable in later regression analyses) included voluntary or involuntary transfers to adult or adolescent, public or private hospital units.

#### 2.4.3. Depression Diagnoses

We used a broad concept of “any Depression” and included all relevant diagnoses from the Mood Disorders (DSM-IV); and the Depressive Disorders and Bipolar and Related Disorders (DSM-5) sections for our classification:

DSM-IV:

Major Depressive Disorder (single or recurrent), Dysthymic Disorder, Organic Mood Disorder, Mood Disorder due to General Medical Condition, Mood Disorder not otherwise specified, Bipolar I Disorder, Bipolar Disorder not otherwise specified, most recent episode depressive, Cyclothymic Disorder.

DSM-5:

Disruptive Mood Dysregulation Disorder, Major Depressive Disorder (single and recurrent episodes), Persistent Depressive Disorder (Dysthymia), Premenstrual Dysphoric Disorder, Substance/Medication-Induced Depressive Disorder, Depressive Disorder Due to Another Medical Condition, Other Specified Depressive Disorder, Unspecified Depressive Disorder.

Bipolar I, Bipolar II Disorder, Unspecified Bipolar and Related Disorder; most recent or current episode depressed.

#### 2.4.4. Analyses

Demographic and clinical characteristics (including any Depression) were presented using descriptive statistics. A post hoc series of comparisons were made based on age stratification (adolescent v young adult) using chi-square (categorical variables) or *t*-tests (continuous variables), to better characterise the two age-based sub-groups.Prevalence of any Depression was expressed as the number and percentage with any Depression diagnosis.The unadjusted magnitude and direction of any Depression diagnosis (no Depression as referent) as a predictor for referral for an inpatient psychiatric admission was calculated using bi-variate logistic regression, with the results expressed as Odds Ratio and 95% Confidence Intervals (OR: CI 95%).A series of sequential multiple logistic regressions models were fitted to estimate any change in association (for any Depression) after adjusting for (each) a priori selected potential confounder (or effect modification) variable (participant characteristics). These results were expressed as adjusted ORs (95% CI).An exploratory multiple logistic regression model best explaining discharge to the psychiatric hospital was developed using a forward elimination technique. The initial selection of variables for inclusion in the step-wise model was based on any demographic or clinical variables that were significantly associated with referral to the psychiatric hospital at the univariate level (see Appendix A). Results were expressed as adjusted ORs (95% CI). We also reported the Nagelkerke’s R^2^, which is an analogue of R^2^ in a linear regression that indicates the proportion of variance explained in the model.

### 2.5. Ethical Approval

This study was conducted in keeping with the rules of the Declaration of Helsinki of 1975 (https://www.wma.net/what-we-do/medical-ethics/declaration-of-helsinki/, accessed on 2 February 2021), revised in 2013 and received Ethical Approval from the Hunter New England Human Research Ethics Committee, Project Number 2020/STE05596. This study and the specific use of HATS clinical case register data, met criteria for a waiver of the individual consent requirements as per Section 2.3.10 of the National Health and Medical Research Council, National Statement on Ethical Conduct in Human Research 2007; updated 2018.

## 3. Results

A total of 1412 patients met the inclusion criteria and two patients’ records were subsequently excluded from the analyses due to missing psychiatric diagnoses data. 

### 3.1. Demographic Characteristics

Of the 1410, (*n* = 654; 46.4%) were adolescents and (*n* = 756; 53.6%) were young adults. Both age groups had a higher proportion of females (68.3% adolescents; 63.0% young adults). Just over half the participants were engaged in some form of work or study. Insecure housing was uncommon and most (76%) were in the middle SEIFA socioeconomic quintile 2–4 (see Table 1).

### 3.2. Prevalence of any Depression and other Psychiatric Diagnoses

Any Depression was diagnosed in over a third (*n*= 500; 35.5%), with a very few Bipolar (most recent episode depressed) diagnoses recorded (*n* = 31: 2.2%). Any Psychotic disorders were infrequent (*n* = 26; 1.8%), whilst Other diagnoses (*n* = 796; 56.5%), any Substance Use Disorder (*n* = 444; 31.5%), any Relational problems (*n* = 553; 39.2%), and any Personality Disorder (*n* = 467; 33.1) were common (see Table 2).

### 3.3. Clinical Characteristics

At the time of psychiatric assessment, two thirds reported no current suicidal ideation or plan (*n* = 941; 66.7%); nearly a fifth had low-moderate suicidal ideation but no active plan (*n* = 256; 18.2%) and one eighth had high suicidal thoughts and/or an active plan (*n* = 175; 12.4%). A high proportion of the cohort reported previous self-harm (*n* = 887; 62.9%). About two-thirds of all participants (*n* = 935; 66.3%) had had contact with psychiatric services in the year before the index presentation to the ED.

The majority of presentations occurred out of business hours (*n* = 1104; 78.3%). One quarter of the participants (*n* = 358; 25.4%) were referred to the psychiatric hospital for a potential admission; with a significantly greater proportion of young adults (*n* = 226; 29.9%) than adolescents (*n* = 132; 20.2%) referred (see Table 2).

### 3.4. Any Depression Predicting Referral to Psychiatric Hospital

The presence of any Depression significantly increased the likelihood of being referred to a psychiatric hospital for admission (OR 3.55; 95% CI 2.77–4.56). When sequentially adjusted for: suicidal ideation, the OR for any Depression was reduced to 2.47 (1.76–3.45); whilst sequential adjustment for age, gender, any Substance Use Disorder, life events (in the past one month), employment/studying, and any psychiatric treatment (in the last 12 months) made little change in the OR for any Depression predicting referral to the psychiatric hospital (see Table 3).

### 3.5. Explanatory Model Predicting Referral to Psychiatric Hospital

In the exploratory explanatory multivariate model, the significant predictors of referral for psychiatric admission were: (increased risk) low/moderate suicidal level (OR 14.27; 95% CI 9.38–21.72), high suicidal level (118.21; 63.23–220.99), depression (2.88; 1.97–4.22), older age (1.12; 1.04–1.21), any psychosis (4.06; 1.15–14.36); and (decreased risk) number of available support persons (0.88; 0.78–0.98). The Nagelkerke’s R^2^ for the entire multivariate model was 0.59, with suicidal level at the time of psychiatric assessment accounting for most of the variance, with a Nagelkerke’s R^2^ of 0.55 (see Table 4).

## 4. Discussion

### 4.1. Prevalence of Depression

The point prevalence of any Depression in our clinical study population (35.5%) was higher than reported rates in the general population (10% for all age groups; 6% for ages 16–24) [28]; and within the wide range identified in the literature for similar clinical populations of hospital-treated deliberate self-poisoning young people (14.6–88%) [10,12]. Young adults and adolescents (37.2% vs. 33.5%) in our study were not significantly different for any Depression prevalence. Since deliberate self-poisoning is a common occurrence in Emergency Department and depression highly prevalent within those clinical populations, we endorse the RANZCP clinical practice guidelines [1], which recommended mental health assessments for all hospital-treated deliberate self-poisonings to identify Depression and other mental illness and for the organisation of clinical services to meet the patients’ needs. Whilst there remains some uncertainties and controversies about the effective identification of and evidence-based treatments for Depression in adolescents and young people [29], there is a recognized requirement in practice guidelines for specialist mental health team availability, primary care watchful waiting for mild cases, specific psychotherapies, and for some pharmacotherapies or combination therapy [30]. Service providers need to plan services, provide budget, and train staff sufficient to meet the needs of this particular population. Non-hospital or community-based interventions, especially those offering services outside of normal business hours, may also be a useful alternative to managing patients with acutely high suicidal thoughts or plans. More recently, 14 community safe spaces (Safe Haven) have become operational across the state of New South Wales, including 1 in our study region, that offer after-hours and weekend peer support for people experiencing high levels of distress and suicidal ideation [31].

### 4.2. Depression Predicting Referral for Psychiatric Hospitalisation

A substantial proportion of those with a clinical diagnosis of any Depression (*n* = 207/500; 41.4%) were referred to the psychiatric hospital (see Appendix A), and any Depression was an independent increased risk factor for referral even after adjustment for relevant potential confounders. Inpatient treatment for serious depression has been recommended under certain conditions. The NICE guidelines suggest *inter alia*; “Inpatient treatment should be considered for children and young people who present with a high risk of suicide, high risk of serious self-harm or high risk of self-neglect, and/or when the intensity of treatment (or supervision) needed is not available elsewhere, or when intensive assessment is indicated” and “When considering admission for a child or young person with depression, the benefits of inpatient treatment need to be balanced against potential detrimental effects, for example loss of family and community support” [30]. Our results are generally consistent with these UK guidelines, although we had no measures of depression severity or the need for high intensity treatment and we did have access to inpatient beds, specialty mental health professionals, and a range of interventions, which are recommend in the NICE guidelines.

### 4.3. Explanatory Model of Referral for Psychiatric Hospitalisation

We developed an exploratory, explanatory model of referral for psychiatric hospitalisation. This model included any Depression (discussed above) and also found: current suicidal level, older (young adult) age group, and any psychosis diagnosis had increased risk; whilst a greater number of available support persons had decreased risk of referral for psychiatric hospitalisation. These results have face validity and plausibly reflect usual clinical practice and the requirements of the NSW Mental Health Act 2007 No 8 [32], where involuntary hospitalisation is used to manage active suicidal thoughts and plans concurrent with serious mental illness (depression and psychosis), especially where social supports are more limited or absent so that a lesser level of care might not be available or appropriate. Adolescents were more likely to be discharged home, when compared to young adults, which could be determined by differential bed availability (more available young adult beds), greater recognition of the potential harms of psychiatric hospitalisation for adolescents or local practice. Recent suggested indications for the psychiatric hospitalisation of children and adolescents focused on three areas: (1) a period of detailed observation to facilitate diagnosis, (2) acute containment of risk and supervised initiation of treatment, and (3) non-psychiatric medical assessment or treatment that cannot, because of the child’s condition, be undertaken in a general medical setting [33,34], and our multivariable model results are broadly consistent with the first two indications.

Inspection of the Nagelkerke’s R^2^ values shows that current suicidal level accounted for the greatest amount of variance in the model. We can only speculate about the clinicians’ motivations for recommending inpatient after-care. It may be that the clinicians were using the suicidal level as a predictor of future suicide or other suicidal behaviours in order to determine inpatient after-care, although it is recognized that any form of risk stratification is too inaccurate to be used to allocate after-care [35,36]. Conversely, the clinicians may have conceptualised current suicidal thoughts and plans as a modifiable risk factor for future suicide, and decided that hospitalisation was necessary to reduce exposure to that risk factor, an approach which has been recommended in clinical practice guidelines for management of deliberate self-harm [1].

Psychiatric hospitalisation is not without potential harms [23], and this should be a consideration for clinicians, especially for young patients and their families. There may also be harms associated with psychiatric hospitalization for care-givers of patients admitted to psychiatric hospital, who experience stigma, disruptions in daily life, worse general mental health, economic strain, and changes in relationships after hospitalisation [37]. Perhaps of most concern is the increased risk of death by suicide associated with psychiatric hospitalisation; estimated in a Danish population study to have an adjusted rate ratio of 44.3 (36.1–54.4) for those admitted to a psychiatric hospital compared with people who had not received any psychiatric treatment [20] and higher than those with non-inpatient mental health care [20]. Rates of suicide can remain high for many years after discharge from psychiatric units, suggested by some authors to indicate a possible long-term negative impact of psychiatric hospitalization [38]. Although much of this increased risk must be due to confounding by indication (the more severely ill are disproportionately admitted to the psychiatric hospital); an Australian study suggested that a proportion of suicides that occur during, or shortly after, psychiatric hospitalisation might be properly regarded as ‘‘nosocomial’’, that is primarily due to factors inherent in hospital-based care [22]. There is evidence to support an increase in risk of death by suicide following discharge from psychiatric units [20], and other harms associated with inpatient psychiatric care [23], but more definitive information would be helpful. Such information would be useful in accurately balancing the benefits and harms of inpatient care when deciding the optimum location of after-care for deliberate self-poisoning. Studies looking at patient experiences and comparing Emergency Department presentations for self-harming or self-poisoning behaviours after discharge from community or inpatient care would be useful in determining the effectiveness of services and would inform the direction of future local service delivery.

Overall, psychiatric hospitalisation is an important alternative for after-care intervention for adolescents and young people presenting to Emergency Department following a deliberate self-poisoning episode. This study has identified potential factors that contribute to psychiatric hospitalisations in a sample presenting to a sentinel unit in Australia. The decision to refer patients for psychiatric admission is complex and further examination of this phenomena is warranted. 

### 4.4. Strengths and Limitations

The use of a retrospective cohort design was appropriate to the study questions. There were few threats to internal validity. The population (selection) biases were minimized by using all index presentations to a regional hospital, with a known referral population for a specified period, where all data are prospectively recorded in a relational database as part of a clinical case register for deliberate self-poisoning. The data entry staff are blind to any study aims. The “exposures” included demographic and clinical characteristics of individual participants, arising directly from the clinical assessment by mental health professionals, and as such are more comprehensive than those variables available in institutional data sets. The “outcome” variable (discharge destination) is also derived from clinical records and there was little or no loss of participants for this short-term outcome. This article was prepared in keeping the STROBE guidelines for reporting cohort studies [39].

All “exposure” and “outcome” variables were derived from clinical assessment and so there may be some measurement error and misclassification. We used clinical assessment for mental health diagnoses and did not use any screeners or diagnostic interview instruments. Socio-economic quintiles were derived from residential post codes. The outcome variable was referral to the psychiatric hospital with a request for inpatient treatment, but we do not know which patients were actually admitted. We used a single multi-variable logistic regression analysis to develop an exploratory, explanatory model of referral to the psychiatric hospital as after-care. We did not test this model in any other data-sets and so there is a possibility that the model is over specified. The results of this study may not necessarily be generalizable to different clinical populations, geographical settings or cultural areas, nor to other forms of self-harm such as cutting. 

## 5. Conclusions

Hospital treated deliberate self-poisoning is common in young persons and any Depression was recognised in over a third of the study population. Depression was an independent risk factor for psychiatric hospitalisation referral, even after adjusting for other potential confounders. Psychiatric inpatient referrals are a commonly utilised intervention for this population and the decision to refer patients to psychiatric units is likely to be complex. Current suicidal level accounted for the largest amount of variance in our model. Though the decision to admit was in accordance with the current guidelines, there is limited evidence on effectiveness of psychiatric hospitalisation in management of deliberate self-poisoning or depression in young people. Services and clinicians working with this population need to take this into account when conducting assessments and providing after-care. The availability of suitable community-based interventions may reduce the need for psychiatric hospitalisation and its potential harms; and the effectiveness and safety of community versus psychiatric inpatient mental health care in these clinical populations for suicidal behaviour outcomes needs to be established. Examining patient experiences of psychiatric hospitalisations and alternate after-care options is also needed to inform the direction of future clinical service delivery.

## Figures and Tables

**Table 1 ijerph-19-15753-t001:** Demographic characteristics by age strata (Adolescents vs. Young Adults).

Categorical		16 to 19 years *N* = 654 *n* (%)	20 to 25 years *N* = 756 *n* (%)	Total *N* = 1410 *n* (%)	Chi Square	*p* Value
Gender	Male	207 (31.7)	280 (37.0)	487 (34.5)	4.5	0.034
	Female	447 (68.3)	476 (63.0)	923 (65.5)		
Cultural background	Indigenous	67 (10.2)	76 (10.1)	143 (10.1)	0.01	0.905
	Non-Indigenous	587 (89.8)	680 (89.9)	1267(89.9)		
Marital status (*missing* = 2)	Single/never married	611 (93.6)	573 (75.9)	1184 (83.9)	81.78	<0.001
	Married/de facto	29 (4.4)	128 (17.0)	157 (11.1)		
	No longer partnered ^1^	13 (2.0)	54 (7.2)	67 (4.8)		
Employment	Employed/Student	346 (52.9)	398 (52.6)	744 (52.8)	1.03	0.597
	Not working/studying	237 (36.2)	287 (38.0)	524 (37.1)		
	Unknown	71 (10.9)	71 (9.4)	142 (10.1)		
Education (highest)	Primary School	42 (6.4)	36 (4.8)	78 (5.5)	60.52	<0.001
	High School	543 (83.0)	518 (68.5)	1061 (75.2)		
	Tertiary	44 (6.7)	143 (18.9)	187 (13.3)		
	Unknown	25 (3.8)	59 (7.8)	84 (6.0)		
Housing	Secure	608 (93.0)	695 (91.9)	1303 (92.4)	1.18	0.553
	Insecure	22 (3.4)	34 (4.5)	56 (4.0)		
	Unknown	24 (3.7)	27 (3.6)	51 (3.6)		
Referral Area	Primary	477 (72.9)	571 (75.5)	1048 (74.3)	1.68	0.433
	Secondary	150 (22.9)	152 (20.1)	302 (21.4)		
	Not in Area	27 (4.1)	33 (4.4)	60 (4.3)		
Socio-economic Quintile (SEIFA) (*missing* = 17)	Lowest (quintile 1)	137 (21.3)	116 (15.5)	253 (17.9)	8.64	0.013
	Middle (quintiles 2–4)	480 (74.5)	591 (78.9)	1071 (76.0)		
	Highest (quintile 5)	27 (4.2)	42 (5.6)	69 (4.9)		
**Continuous**		**16 to 19 years** **Mean (SD)**	**20 to 25 years** **Mean (SD)**	**Total** M**ean (SD)**	** *t* ** **-Test**	** *p* ** **-Value**
Age at presentation (years)		17.72 (1.03)	22.11 (1.65)	20.07 (2.60)	−59.02	<0.001
Length of stay (hours)		19.85 (17.90)	22.80 (33.35)	21.45 (27.32)	−2.13	0.033

^1^ Due to divorce, separation, or death.

**Table 2 ijerph-19-15753-t002:** Clinical characteristics by age strata (Adolescents vs. Young Adults).

Clinical Characteristics		16 to 19 years *N* = 654 *n* (%)	20 to 25 years *N* = 756 *n* (%)	Total *N* = 1410 *n* (%)	Chi Square	*p* Value
DSM Diagnostic Categories ^1,2^	Any Depression	219 (33.5)	281 (37.2)	500 (35.5)	2.08	0.149
	Depressive Disorder	211 (32.3)	258 (34.1)	469 (33.3)	0.55	0.459
	Bipolar Disorder	8 (1.2)	23 (3.0)	31 (2.2)	5.40	0.020
	Any Anxiety	114 (17.4)	97 (12.8)	211 (15.0)	5.83	0.016
	Any Psychosis	8 (1.2)	18 (2.4)	26 (1.8)	2.60	0.107
	Any Substance Use	157 (24.0)	287 (38.0)	444 (31.5)	31.66	<0.001
	Any Relational Problem	290 (44.3)	263 (34.8)	553 (39.2)	13.43	<0.001
	Any Personality Disorder	207 (31.7)	260 (34.4)	467 (33.1)	1.19	0.276
	Other	387 (27.4)	409 (29.0)	796 (56.5)	3.34	0.068
Suicidal Level ^3^	None	457 (69.9)	484 (64.0)	941 (66.7)	6.74	0.081
	Low/Moderate ^3a^	111 (17.0)	145 (19.2)	256 (18.2)		
	High ^3b^	68 (10.4)	107 (14.2)	175 (12.4)		
	Unknown	18 (2.8)	20 (2.6)	38 (2.7)		
Previous self-harm	Absent	217 (33.2)	306 (40.5)	523 (37.1)	7.81	0.005
	Present	437 (68.7)	450 (61.5)	887 (62.9)		
Psychiatric Treatment (past 12 months)	No contact	211 (32.3)	264 (34.9)	475 (33.7)	4.91	0.086
	Outpatient only	329 (50.3)	337 (44.6)	666 (47.2)		
	Inpatient	114 (17.4)	155 (20.5)	269 (19.1)		
Life Events (past month) (missing = 19)	None	219 (34.1)	273 (36.5)	492 (34.9)	0.95	0.62
	One	244 (37.9)	270 (36.1)	514 (36.5)		
	Two or more	180 (28.0)	205 (27.4)	385 (27.3)		
Presentation time	In hours	151 (23.1)	155 (20.5)	306 (21.7)	1.38	0.240
	Out of hours	503 (76.9)	601 (79.5)	1104 (78.3)		
**Continuous**		**16 to 19 years** **Mean (SD)**	**20 to 25 years** **Mean (SD)**	**Total** **Mean (SD)**	***t*-test**	***p*-value**
No of diagnoses (missing 6)		1.26 (0.85)	1.39 (0.82)	1.33 (0.83)	−2.89	0.004
No of support persons (missing 211)		3.21 (2.44)	2.78 (1.39)	2.98 (2.18)	3.39	0.001
No of life events (past month) (missing 19)		1.06 (1.04)	1.05 (1.08)	1.06 (1.06)	0.13	0.897
**Outcome**		**16 to 19 years** ***n* (%)**	**20 to 25 years** ***n* (%)**	**Total** ***n* (%)**	**Chi square**	***p* value**
Discharge destination	Psychiatric inpatient referral	132 (20.2)	226 (29.9)	358 (25.4)	17.46	<0.001
	Other detination ^4^	522 (79.8)	530 (70.1)	1052 (74.6)		

^1^ Diagnostic and Statistical Manual DSM-IV or DSM -5 depending on coding used at presentation. ^2^ More than one diagnosis is possible. ^3^ At the time of psychiatric assessment. ^3a^ Some thoughts with no plan. ^3b^ Intense thoughts and/or plan. ^4^ Home/usual residence or police custody or self-discharged from general hospital.

**Table 3 ijerph-19-15753-t003:** Unadjusted and adjusted estimates for Any Depression predicting referral for psychiatric admission.

**Unadjusted**	**Discharge to Psychiatric Hospital** **OR (95% CI)**
Any Depression	3.55 (2.77–4.56)
**Adjusted (sequentially) for**	**AOR (95% CI)**
Age	3.54 (2.75–4.54)
Gender	3.64 (2.83–4.68)
Suicidal Level	2.47 (1.76–3.45)
Substance Use	3.69 (2.84–4.75)
Life events (past month)	3.57 (2.78–4.59)
Employed/studying	3.66 (2.84–4.71)
Psychiatric treatment (past 12 months)	3.55 (2.75–4.58)

**Table 4 ijerph-19-15753-t004:** About here: Univariate and multivariate analysis of variables predicting referral for psychiatric admission.

Predictor Variable	Univariate Analysis		Multivariate Analysis	
	OR (95% CI)	Unadjusted Individual Nagelkerke R^2^	OR (95% CI)	Adjusted Cumulative Nagelkerke R^2^
Suicidal Level				
None	Referent	0.549	Referent	0.551
Low/Moderate	16.31 (11.32–23.48)		14.27 (9.38–21.72)	
High	146.88 (83.19–259.34)		118.21 (63.23–220.99)	
Unknown	9.22 (4.52–18.79)		14.91 (5.69–39.06)	
Any Depression	3.55 (2.77–4.56)	0.103	2.88 (1.97–4.22)	0.572
Age on admission (continuous)	1.1 (1.05–1.15)	0.018	1.12 (1.04–1.21)	0.582
No. support persons (continuous)	0.84 (0.77–0.91)	0.026	0.88 (0.78–0.98)	0.586
Any Psychosis	3.01 (1.38–6.56)	0.008	4.06 (1.15–14.36)	0.590

## Data Availability

The data for this study are not publicly available. Data access can be requested from the Corresponding Author.

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
