# Peer review of "Prevalence of Depression and Predictors of Discharge to a Psychiatric Hospital in Young People with Hospital-Treated Deliberate Self-Poisoning at an Australian Sentinel Unit"

_ijerph, 2022, doi:10.3390/ijerph192315753_

Round 1

Reviewer 1 Report

In medical sciences, there is a traditional way to write reports. The document and writing style is trained constantly till there is no other way in which you can write. All physical practitioners are reporting this way and process data through SPSS. The IJERPH journal is meant for a much broader audience, containing many people without that basic training. Clearly that will make a difference for people that want to publish in IJERPH, not only in manuscript structure as also in the writing style.

Depression is used as a singular cause for health issue. However, over the recent years the understanding has grown that Depression comes never in isolation. The authors recognize that in “future research”, leaving the issue whether comorbidity is an important factor or not. Unquestionable, Depression comes as a social parameter and therefore health can also be seen as a parameter of depression in a social study instead of vice versa. This brings the question whether the proposed text is too limited.

There is an abundance of information in the manuscript. Too much for comfort, as the reader has to decide on the relevance of each fact. The subsection “strengths and limitations” could bring help but again truths beyond what is automatically given by SPSS are scarce. Consequently, it is hard to find the few facts of most relevance. The medical report expects the writers to refrain from a personal interpretation; the scientific report has to provide personal interpretations ánd their substantiation.

Author Response

We agree with the reviewer that the report is highly technical and the report is written in a way to reflect that. This is probably not a paper for the general readership of the IJERPH who do not have that training or those technical skills. We have followed the manuscript requirements of the journal required by IJERPH. 

The mental health comorbidities are reported in Table 2 and in the text 3.2.
The specific aims of the study were to estimate the association(s) of Depression with discharge destination.

We believe the points raised in the Discussion and Conclusion are supported by the results and do not require any further revision.

Reviewer 2 Report

Thank you for the opportunity to review this interesting manuscript. Authors represented in detail the prevalence of depression in young adults, and analysed predictors of discharge to a psychiatric hospital after intentional self-intoxication. Introduction and methods are written in detail, and the aims are clearly represented. Results and discussion follow the subheading presented in the aims.

The following comments/questions are minor, and are intended to further improve the manuscript.

1)     Authors should consider adding the STROBE check list for cohort studies (future readers might find it important for critical appraisal or interpretation)

2)     Paragraph 2.4.2. Clinical Characteristics:

     Other chronic conditions/diagnoses were not collected? (for instance other medical conditions participants could have that could have influenced any depression diagnoses, such as multiple sclerosis, physical disabilities…)

3)     Can you comment on not including the variable housing and education in the multiple logistic regression model (variables were significant in the univariate model)? Did such a model account for less variance?

4)     You have looked at a time period of 10 years. Do you have any data whether those who declined referral were re-hospitalised for self-harm again, or that those who were referred had less re-hospitalisations? Or even vice-versa since young patients are sensitive to psychiatric hospitalisations?

5)     Have you considered including other forms of self-harm in the model?

Minor comments regarding style of manuscript (if authors find appropriate) to increase ease of reading:

·        Consider writing paragraphs 2.4.1. -2.4.3. in bullet point or tables (readers might find it strenuous to read all the variables and its options as a full sentence)

Consider shortening the conclusion 

Reviewer 3 Report

Introduction

A study hypothesis is missing

Discussion and conclusions

"While there is evidence to support an increase in risk of death by suicide following discharge from psychiatric units, there is not much information about other harms associated with inpatient psychiatric care."

After this statement, a reference is needed.

Such concluding statements are too strong for the strength of a cohort study. Please soften statements regarding the usefulness of psychiatric hospitalization after suicidal attempts and the referral to stigma as a consequence of psychiatric hospitalization, it seems more like a statement emphasizing stigma rather than a scientific finding.

Suggested references

Pompili M, Iliceto P, Luciano D, Innamorati M, Serafini G, Del Casale A, Tatarelli R, Girardi P, Lester D. Higher hopelessness and suicide risk predict lower self-deception among psychiatric patients and non-clinical individuals. Riv Psichiatr. 2011 Jan-Feb;46(1):24-30. PMID: 21443138.
